# Localized Infections with *P. aeruginosa* Strains Defective in Zinc Uptake Reveal That Zebrafish Embryos Recapitulate Nutritional Immunity Responses of Higher Eukaryotes

**DOI:** 10.3390/ijms24020944

**Published:** 2023-01-04

**Authors:** Valerio Secli, Claudia Di Biagio, Arianna Martini, Emma Michetti, Francesca Pacello, Serena Ammendola, Andrea Battistoni

**Affiliations:** 1Department of Biology, University of Rome Tor Vergata, Via della Ricerca Scientifica, 00133 Rome, Italy; 2Laboratory of Experimental Ecology and Aquaculture, Department of Biology, University of Rome Tor Vergata, 00133 Rome, Italy; 3Council for Agricultural Research and Economics, Research, Centre for Animal Production and Aquaculture, Via Salaria 31, 00015 Monterotondo, Italy

**Keywords:** nutritional immunity, zinc transport, zebrafish, *P. aeruginosa*, host–pathogen interaction

## Abstract

The innate immune responses of mammals to microbial infections include strategies based on manipulating the local concentration of metals such as iron (Fe) and zinc (Zn), commonly described as nutritional immunity. To evaluate whether these strategies are also present in zebrafish embryos, we have conducted a series of heart cavity-localized infection experiments with *Pseudomonas aeruginosa* strains characterized by a different ability to acquire Zn. We have found that, 48 h after infection, the bacterial strains lacking critical components of the Zn importers ZnuABC and ZrmABCD have a reduced colonization capacity compared to the wild-type strain. This observation, together with the finding of a high level of expression of Zur-regulated genes, suggests the existence of antimicrobial mechanisms based on Zn sequestration. However, we have observed that strains lacking such Zn importers have a selective advantage over the wild-type strain in the early stages of infection. Analysis of the expression of the gene that encodes for a Zn efflux pump has revealed that at short times after infection, *P. aeruginosa* is exposed to high concentrations of Zn. At the same time, zebrafish respond to the infection by activating the expression of the Zn transporters Slc30a1 and Slc30a4, whose mammalian homologs mediate a redistribution of Zn in phagocytes aimed at intoxicating bacteria with a metal excess. These observations indicate that teleosts share similar nutritional immunity mechanisms with higher vertebrates, and confirm the usefulness of the zebrafish model for studying host–pathogen interactions.

## 1. Introduction

Among the mechanisms of innate immunity with which higher eukaryotes defend themselves against infections, a particularly interesting role is played by a series of strategies indicated by the term nutritional immunity. The latter involves the modulation of metal availability for the microorganisms triggering the infection [1]. Nutritional immunity includes strategies to either decrease the availability of metals such as iron (Fe), zinc (Zn), manganese (Mn), nickel (Ni), and copper (Cu) from mucosal sites or in serum, or the use of high concentrations of metals such as Zn and Cu in vesicular compartments of phagocytes to poison invading microorganisms [2]. Pathogenic bacteria and fungi have developed diversified and often sophisticated strategies to evade these antimicrobial responses. These bacterial responses favor the acquisition of metals even in nutrient-poor environments, or allow the escape from intoxication mechanisms caused by high concentrations of metals [3,4].

*Pseudomonas aeruginosa* is an opportunistic pathogen capable of colonizing metal-poor environments via a variety of effective metal acquisition strategies. This bacterium can proliferate in iron-poor environments by producing siderophores, such as pyoverdine and pyochelin [5,6], and can cope with Zn deficiency by expressing multiple import systems with high affinity for this metal [5]. These include the ZnuABC transporter, common to most Gram-negative bacteria [6,7], and the pseudopaline synthesis and transport system encoded by the *zrmABCD* operon [5,8]. Pseudopaline is a small organic molecule released by the bacterium in conditions of Zn deficiency [5,8]. It captures this metal in the extracellular space, also tearing it from other Zn-binding proteins [9]. Similar metallophore-based Zn acquisition systems have also been identified in a limited number of different bacteria [10], including the human pathogens *Yersinia pestis* [11,12] and *Staphylococcus aureus* [13,14]. This marked ability to resist the mechanisms of nutritional immunity may provide an essential contribution to *P. aeruginosa* infection ability. In this regard, it is useful to stress that *P. aeruginosa* is responsible for severe debilitating infections in immunocompromised individuals, and is also the most common bacterium isolated from chronic wounds [15]. Furthermore, it is the predominant opportunistic pathogen identified in Cystic Fibrosis (CF), where it can lead to chronic, almost ineradicable lung infections [16,17]. Several animal models have been developed to study the interaction of *P. aeruginosa* with its host, primarily aimed at analyzing infections in the context of CF [18,19]. When deprived of a functional Cystic Fibrosis Transmembrane Regulator (CFTR) channel, large animals, such as pigs and ferrets, faithfully recapitulate many aspects of the human disease. However, managing these animals on a laboratory scale is complex [20,21,22]. Mouse models are more frequently used, although the inactivation of the murine CFTR gene does not result in an infection-independent inflammation comparable to that observed in human patients [23,24]. Therefore, considering all the scientific, practical, economic, and ethical issues in using mammalian models, there has been an interest in exploring the possible use of alternative animal models. Among these, zebrafish are regarded with much interest. Zebrafish have an excellent potential as a vertebrate model for studying host–pathogen interactions, particularly for investigating different aspects of bacterial infections during CF [25,26,27]. So far, no studies have investigated whether zebrafish possess nutritional immunity mechanisms comparable to those of higher organisms. To seek answers to this issue, we have carried out this study to evaluate whether zebrafish can control *P. aeruginosa* infections through mechanisms involving the manipulation of Zn availability.

## 2. Results

### 2.1. Development of an Experimental Model of P. aeruginosa Infection in Zebrafish Embryos

The infectious ability of *P. aeruginosa* was investigated in zebrafish embryos at 48 h post fertilization (hpf), a stage in which the embryos are already equipped with primitive macrophages capable of fighting infections through phagocytosis mechanisms [28,29,30]. Figure 1 shows the survival curves of zebrafish embryos infected with the PA14 wild-type strain, indicating that mortality increases as a function of the infectious dose (50 < 2000 < 4000 CFU). It can be noted that the mortality rate is not directly proportional to the increase in the number of microinjected bacteria, and that survival is high even when the embryos are inoculated with the highest dose of bacteria. Mortality reaches a plateau at 48 h post-infection (hpi). Microinjection does not cause mortality, as indicated by the control group in which only PBS was microinjected.

### 2.2. Mixed Infections Reveal a Contribution of Zn Uptake Systems to P. aeruginosa Virulence in Zebrafish

Since the mortality induced by localized pericardial cavity (PC) infections is low, this mode of infection is not suitable for analyzing subtle differences in the colonization capacity of the wild-type versus mutant *P. aeruginosa* strains. Therefore, competition assays were employed to analyze the possible contribution of the genes responsible for Zn acquisition to the infectious ability of this pathogen. For these assays, an infectious dose of 2000 CFU was employed. Competition between the wild-type strain and the *zrmA* strain revealed that deletion of the pseudopaline receptor ZrmA has a poor impact on the survival of *P. aeruginosa* in this host (Figure 2A). However, strains with a substantially reduced ability to import Zn, such as the *znuA* or the *znuAzrmA* mutants, were outcompeted by the wild-type strain at 24 and 48 hpi (Figure 2B,C). It is interesting to observe that a contribution of the pseudopaline-based Zn uptake system was revealed by the competition between the *znuAzrmA* and the *znuA* strains (Figure 2D). Indeed, in this experiment, we observed a clear advantage of the *znuA* mutant over the *znuAzrmA* strain at 48 h post-infection (Figure 2D). These results suggest that the ability to import Zn is critical for *P. aeruginosa* during the infection of zebrafish embryos.

### 2.3. The Early Response of Zebrafish to P. aeruginosa Infections Involves Zn Intoxication

Despite competition experiments disclosing a clear disadvantage of the *znuA* and *znuAzrmA* mutants at 48 hpi, this defect was not always evident in the short-term response. In fact, at 6 hpi, the single *znuA* or *zrmA* mutant showed a statistically significant advantage over the wild-type strain (Figure 2B). Similarly, at this time point, the *zrmAznuA* mutant is advantaged over the *znuA* mutant. These observations suggest that bacteria with an impaired ability to import Zn could be somehow advantaged in the early stages of the infection. This finding triggered the speculation that the host copes with the infection by exploiting the ability of phagocytes to intoxicate bacteria with Zn during the early stages of the infection. To test this hypothesis, we investigated whether the expression levels of genes belonging to the ZnTs family of Zn transporters were altered in bacterial infections. We focused on the genes *slc30a1a*, *slc30a4*, *slc30a5*, and *slc30a7*, which encode for putative Zn transporters potentially involved in Zn efflux, as well as in the compartmentalization of the metal in intracellular organelles [31,32]. Zebrafish embryos were infected with *P. aeruginosa* wild-type, and mRNA was extracted from fish homogenates. RT-PCR analyses showed that bacterial infection significantly increased the expression of *slc30a1a* and *slc30a4* already after 6 hpi, while *slc30a5* and *slc30a7* did not show a significant response (Figure 3).

### 2.4. P. aeruginosa Adapts to Zebrafish by Modulating the Expression of Distinct Zn-Responsive Genes at Different Stages of Infection

Bacteria can cope with Zn intoxication by activating efflux pumps that maintain intracellular Zn concentration below a critical threshold. To verify whether this mechanism is active during the infection of zebrafish embryos, the expression levels of the gene encoding the *P. aeruginosa* metal-transporting P-type ATPase (*PA14-16660*) were evaluated by RT-PCR during the infection. Transcription of *PA14_16660* significantly increased at 6 hpi compared to that in bacteria grown in LB. Yet, its expression decreased at 24 hpi and was undetectable at 48 hpi (Figure 4A). On the contrary, Zur-regulated genes responding to Zn shortage (*zrmA*, *dskA2*, and *rpmE2*) [33] were poorly expressed at 6 hpi and 24 hpi (Figure 4B–D), but were upregulated in the late stage of infection (48 hpi, Figure 4B–D). Taken together, these results support the hypothesis that the early response of zebrafish to *P. aeruginosa* infection occurs via mechanisms of Zn intoxication, while at longer times the surviving bacteria face Zn starvation.

## 3. Discussion

Previous investigations have demonstrated the critical role of Zn in the ability of *P. aeruginosa* to interact with its hosts. Inactivation of known high-affinity Zn acquisition systems negatively affects the ability of this pathogen to express virulence features and cause disease in infected animals. It was shown that deletion of *znuABC* impairs *P. aeruginosa* resistance to Calprotectin, reduces alginate production, and the activity of extracellular Zn-containing proteases, including LasA, LasB, and protease IV, and decreases the ability of *P. aeruginosa* to spread during systemic infections [6]. These defects are exacerbated in a *znuABC* and *zrmABCD* deficient strain, which exhibits a further reduction in the ability to release Zn-dependent extracellular proteases and pyoverdine, has swarming and swimming motility defects, and has reduced capacity of forming biofilms. Moreover, the impairment of these Zn uptake systems drastically affects *P. aeruginosa* ability to cause acute lung infections in mice [5]. 

Different studies have exploited zebrafish as an infection model to investigate host–pathogen interactions [30,34,35,36]. Among the most valuable advantages of zebrafish are the optical transparency of the early stages, the rapid development, and the possibility to investigate immune responses [36]. In this regard, it should be highlighted that the development of adaptive immunity is completed only after about 4 weeks post-fertilization. Therefore, the defense against infections in the embryonal state is ensured exclusively by the innate immune system [37]. Although different studies have recently reviewed the usefulness of zebrafish for shedding light on fundamental concepts underlying vertebrate immune response to infections by microbial pathogens [37,38], it is not yet known whether the response of zebrafish to infections involves nutritional immunity mechanisms. To test this possibility, we have investigated the capability of this basal teleost to contain *P. aeruginosa* infections by manipulating Zn availability. To this aim, zebrafish embryos were challenged with a local infection via injections of the pathogen into the pericardial cavity. For this first investigation, the choice of the injection site and the consequent generated local infection were aimed at increasing the survivorship of the zebrafish embryos, since systemic infections result in significantly lower survival rates [39]. Furthermore, local infections were shown to induce a strong host defense [34], which represents the most suitable conditions for investigating whether manipulation of Zn concentrations is among the protective strategies of zebrafish.

The infections were carried out with wild-type *P. aeruginosa* and mutant strains defective for Zn acquisition, i.e., the single mutants *znuA* and *zrmA*, and the double mutant *znuAzrmA*. The embryos showed great resistance to infection at the PC site with the wild-type *P. aeruginosa* strain. In fact, by infecting the embryos with 50 CFU, we only observed a reduction in the survival rate of about 7% after more than 40 hpi. However, even infections with very high doses (2000 CFU and 4000 CFU) showed a reduction in the survival rate between 13% and 16%. Our observations align with previous studies conducted with *S. aureus* showing a great resistance of zebrafish to infections in the pericardial cavity [39]. Indeed, zebrafish local cavities, including the PC, are hypothesized to be rich in immune cells (macrophages and neutrophils) that contribute to fighting bacterial proliferation. Furthermore, in contrast to the yolk sac, such regions are less rich in nutrients that can be exploited by invading bacteria [40]. 

Competition experiments highlighted that *P. aeruginosa* strains with a reduced ability to acquire Zn (*znuA* or *znuAzrmA* mutants) are outcompeted by the wild-type strain at 24 hpi and 48 hpi. Our observations agree with previous results showing a reduced virulence in mice of the single mutants *znuA* and *zrmA*, suggesting that both the ZnuABC transporter and pseudopaline contribute to Zn uptake in infected hosts [5]. However, competition experiments suggested a more complex relationship between zebrafish and *P. aeruginosa*, not solely based on essential metal deprivation mechanisms. The analyses carried out at 6 hpi showed a competitive advantage of the single *znuA* or *zrmA* mutant, compared to the wild-type strain. This finding indicates that in the early stages of infections, the inactivation of each of the two major high-affinity Zn import systems favors bacterial survival in the host. These observations may agree with a series of studies conducted in recent years, which have shown that the host can exploit the toxicity of metals to poison intracellular bacteria directly [2]. Several studies have documented fluctuations in Zn concentrations within immune cells in response to infections or microbial products [41,42,43]. For example, macrophages mobilize and traffic Zn to phagosomes and intracellular compartments that co-localize with various bacterial pathogens, including *Mycobacterium tuberculosis*, *Salmonella enterica*, and *Escherichia coli*. In a study by Stocks and colleagues, co-localization of Zn vesicles was observed with an *E. coli* strain that specifically reports Zn stress [43]. It has also been observed that Zn compartmentalization within phagosomes can be mediated by a member of the Zn transporter family ZnT (SLC30) or via fusion with Zn-containing vesicles (“zincosomes”) [41]. Zn accumulation in lysosomes and azurophilic granules has also been observed in neutrophils, showing that the resistance of phagocytosed *Streptococcus pyogenes* is mediated by Zn efflux systems [44]. Ultimately, these studies, coupled with the evidence that high Zn levels cause severe stress, suggest that innate immune cells can use this metal to poison intracellular bacteria. 

To support the hypothesis that the competitive advantage showed by strains lacking Zn importers during the early stages of infection can be explained by the ability of zebrafish to contrast bacterial infections through Zn intoxication mechanisms operated by phagocytes, we analyzed the expression of a group of zebrafish genes encoding for putative Zn transporters. Those genes are homologous to the ones (SLC30A1, SLC30A4, SLC30A5, and SLC30A7) that, in mammals, mediate the export of Zn ions from the cytoplasm into vesicles or compartments containing bacteria. For example, it has been shown that the expression of SLC30A1 is induced by LPS in primary human macrophages, with its ectopic expression in THP-1 monocytic cells sufficient to drive the formation of Zn vesicles and subject intracellular *E. coli* to Zn stress [41,45]. It has also been observed that the interleukin IL-4 can upregulate SLC30A4, causing an increase in labile Zn within the phagolysosome, triggering mechanisms that can modulate the intracellular survival of pathogens [46]. In contrast, the SLC30A5 and SLC30A7 transporters are known to play a role in the sequestration of Zn within the Golgi apparatus in mouse macrophages. This ability to compartmentalize Zn has been associated with a macrophage response to fungal pathogens, rather than bacterial infections [47]. Our results (Figure 3) show that the expression levels of *slc30a1a* and *slc30a4* are significantly increased in zebrafish upon *P. aeruginosa* infection, suggesting that they have a role in controlling infections through mechanisms involving Zn redistribution in phagocytes, as observed in mammals. 

At the same time, we analyzed the expression of bacterial genes involved in *P. aeruginosa* responses to Zn excess or deficiency conditions (Figure 4). These analyses revealed that *PA_16660* is highly expressed during the early phases of infection. This is a gene encoding a metal-transporting P-type ATPase homologous to the ZntA protein, known to protect bacteria such as *M. tuberculosis*, *E. coli*, and *S. enterica* serovar Typhimurium from phagocyte-mediated Zn intoxication [41,42]. In contrast, the analysis of the Zur-regulated genes *zrmA*, *dskA2*, and *rpmE2* revealed that genes involved in the adaptation of bacteria to Zn-deprived conditions are repressed in the early stages of infection, but highly expressed at 48 hpi. These observations are in very good agreement with the hypothesis that in the early stages of infection, *P. aeruginosa* localizes in environments characterized by a high concentration of Zn. In contrast, at a later stage, the bacteria localize in host niches characterized by poor Zn availability. This picture suggests that following PC-localized infections, bacteria are rapidly engulfed in phagocytes that counter the infection by mechanisms involving poisoning with high concentrations of Zn. Subsequently, the bacteria capable of surviving this antimicrobial strategy localize in environments where the host controls bacterial proliferation using mechanisms based on the sequestration of the metal. In apparent contrast to the observation that in the late stages of infection the bacteria are under zinc starvation, gene expression data revealed elevated transcription of *slc30a1a* and *slc30a4* even at 48 hpi. Previous studies have demonstrated that *slc30a1a* and *slc30a4* are induced by LPS in human monocyte-derived macrophages, in mouse bone marrow-derived macrophages, and in murine dendritic cells [40,48], suggesting that this is a conserved regulatory mechanism in vertebrates. Since at 48 hpi the infection has not yet resolved, we can hypothesize that the bacteria still present in the zebrafish body release LPS that maintain an elevated ability of macrophages to kill bacteria through zinc intoxication. LPS may also promote a more general redistribution of zinc throughout the body, including a reduction in the plasma concentration of the metal, thus contributing to conditions of zinc deficiency outside macrophages [49].

Overall, the antimicrobial mechanisms suggested here, based on Zn intoxication or Zn sequestration, recapitulate the general mechanisms of nutritional immunity to infections already observed in higher organisms. These results confirm the importance of Zn in the host–*Pseudomonas* interaction. They indicate that zebrafish are an interesting model for assessing the contribution of metal transport systems to the ability of pathogens to colonize their hosts. Some studies have already identified zebrafish as a potential model for preclinical studies on microbial infections in Cystic Fibrosis [24,25,26]. Our findings reveal further elements of similarity between the immune response of teleosts and higher vertebrates, and confirm that the zebrafish is a potentially useful model for this purpose. Zebrafish embryos are already widely employed for toxicology and chemical screenings, providing a suitable tool for quick and large-scale drug testing. This study confirms that zebrafish embryos offer a cost-effective, straightforward methodology for exploratory trials on host–pathogen interactions and preliminary screenings. Lastly, besides the advantages of a localized infection model versus a systemic infection, the employed method was meant to provide a cost-effective and easy procedure for pathogen delivery that does not require specialized equipment or highly skilled staff. 

## 4. Materials and Methods

### 4.1. Bacterial Strains and Growth Conditions

Bacterial strains used in this work are *P. aeruginosa* PA14 and its mutant derivatives defective in zinc uptake: *znuA*, which lacks the periplasmic component of the ZnuABC high-affinity zinc uptake system, *zrmA*, lacking the TonB-dependent outer membrane receptor for pseudopalin, and *znuAzrmA*, combining the two mutations (Table 1). Bacteria were grown by streaking glycerol stocks on *Pseudomonas* Isolation Agar (PIA; Becton Dickinson, Franklin Lakes, NJ, USA) supplemented with 0.1 g L^−1^ gentamicin (Sigma Aldrich, St Louis, MO, USA) when needed, and incubated overnight at 37 °C. The strains were routinely grown in LB broth (Bactotryptone 10 g L^−1^, yeast extract 5 g L^−1^, NaCl 10 g L^−1^) at 37 °C under shaking. Bacterial concentration was estimated by measuring the optical density at 600 nm, and adjusted to appropriate Colony Forming Units/nL (CFU/nL) in sterile PBS containing 0.5% Phenol Red tracking dye (Sigma Aldrich, St Louis, MO, USA) to assist in the visualization of injection.

### 4.2. Zebrafish Embryos Infection

Zebrafish (*Danio rerio*) rearing was conducted at the Experimental Biology and Aquaculture Laboratory, University of Rome ‘Tor Vergata’, Italy. The water was processed with an osmotic filtering system (4 Stages Pro System 75GPD, Askoll, Vicenza, Italy), supplemented with salts (0.005% Sera Mineral Salts, 1 mM NaHCO_3_), and UV sterilized (External Sterilizer Pro 18W, AQL, Bari, Italy). Zebrafish embryos of the AB line were incubated in 500 mL of E3 medium [50] supplemented with 0.0002% methylene blue in a thermostatic chamber at 28 °C, with 14:10 light-dark photoperiod. Vital eggs were staged at 48 hpf according to Kimmel et al. [51], manually dechorionated and anesthetized in tricaine methanesulfonate (MS-222, 50 μg mL^−1^) (Western Chemicals, Inc., Ferndale, WA, USA) before microinjection. Bacteria, grown overnight in LB broth at 37 °C with aeration, were diluted in sterile PBS and microinjected (5 nL per embryo) into the pericardial cavity (PC) with the Nanoject II Auto-Nanoliter Injector (Drummond Scientific Company, Broomall, PA, USA) under a stereomicroscope (Axiozoom V.16, Zeiss, Jena, Germany) to ascertain the success of the infection visually. To verify that the microinjection procedure did not influence the zebrafish survival, a group of embryos was injected with PBS alone and used as a control group. Embryos were returned to E3 medium, incubated at 28 °C, and monitored for survival at regular intervals. Living embryos were identified by monitoring the presence of heartbeat, motility capacity, and blood circulation under a stereomicroscope. All the experiments on zebrafish were performed within 5 days of fertilization, respecting all the required Italian and European parameters for animal experiments without the need for ethical approval.

### 4.3. Competition Assay in Zebrafish Embryos

Embryos at 48 hpf were microinjected with 2000 CFU of a 1:1 mixture (input) of two different PA14 strains (strain A and strain B). The input ratio (strain A/strain B) was confirmed by plating an aliquot of the mixture on PIA plates and replica plating 200 colonies on PIA–gentamicin plates. Bacteria were recovered from infected embryos following a previously described protocol with minor modifications [28]. At 6, 24, and 48 h post-infection (hpi), viable embryos were randomly divided into groups of 5 animals. The embryos were euthanized with an overdose of MS-222 (0.3 g L^−1^), rinsed twice in E3 medium, and homogenized in 0.5 mL E3 medium with a micropestle (Sigma Aldrich). After homogenization, 0.2 mL PBS and 0.13 mL Triton X-100 1% were added to each sample and vortexed for 1 min. Serial dilutions of homogenates (outputs) were plated on PIA and incubated overnight at 37 °C. The next day, at least 200 colonies from each group were replica plated on gentamicin-supplemented PIA plates to evaluate the ratio (strain A/strain B) in the outputs. Each competitive index (CI) was calculated using the formula CI = output (strain A/strain B)/input (strain A/strain B). 

By this formula, CI > 1 for strain A outcompeting strain B, CI < 1 for strain B outcompeting strain A, and CI = 1 for a strain A/strain B equal ratio in the input and in the output (same fitness of the two strains).

### 4.4. RNA Extraction and Real Time-PCR

Each embryo was infected with 4000 CFU of PA14 wild-type strain for RT-PCR experiments. RNA was isolated from a pool of 10 homogenized animals using the TRIzol Reagent (Invitrogen, Waltham, MA, USA), following a protocol already described [52]. The RNA from an overnight culture of *P. aeruginosa* PA14 wt was extracted with the RNAeasy kit (Qiagen, Hilden, Germany), according to the manufacturer’s protocol, with the addition of DNase (Qiagen) and lysozyme (Sigma Aldrich). RNA concentration was determined with a NanoDrop™ Lite Spectrophotometer (Thermo Fisher Scientific, Waltham, MA, USA). From each sample, 1 µg of RNA was reverse transcribed with the PrimeScript RT Reagent Kit and gDNA Eraser (Takara Bio Inc., Shiga, Japan). The primers used for RT-PCR were designed using Primer3 [53] and are listed in Table 2. RT-PCR reactions were performed in triplicate in 10 µL reaction mixtures containing: cDNA 50 ng, primers 0.3 µM, and SYBR green 50% (PowerUp SYBR Green Master Mix, Thermo Fisher). Amplifications were performed in a Thermo Fisher (QuantStudio3) thermocycler with the following parameters: (i) initial denaturation at 95 °C for 4 min; (ii) 40 cycles of denaturation at 95 °C for 20 s, primer annealing at 60 °C for 30 s and extension at 72 °C for 30 s; (iii) melting curve, from 50 to 90 °C (rate: 0.58 °C every 5 s). The mRNA fold induction was calculated using the DDCt method [54] and normalized to the following housekeeping genes: *oprI* and/or *rpoD* for *P. aeruginosa* and *ef1α* for zebrafish.

### 4.5. Statistical Analyses

All statistical analyses were conducted using GraphPad Prism v.8.3.1 software. Student’s *t*-test, One-way ANOVA with Bonferroni’s correction for multiple testing, or Two-way ANOVA with Tukey’s multiple comparison test were calculated as specified in the figure captions.

## Figures and Tables

**Figure 1 ijms-24-00944-f001:**
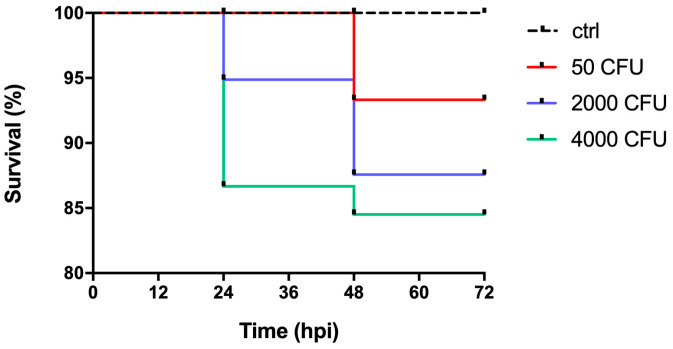
Zebrafish survival to *P. aeruginosa* infections. Kaplan–Meier survival curves of zebrafish embryos microinjected in the pericardial cavity as described in Section 4.2, with increasing doses of PA14 wild-type or PBS alone (ctrl). The survival was calculated as a percentage of the total injected embryos (*n* = 15) for each treatment group and was monitored at post infections time intervals (hpi). Data were obtained from one experiment representative of three independent experiments.

**Figure 2 ijms-24-00944-f002:**
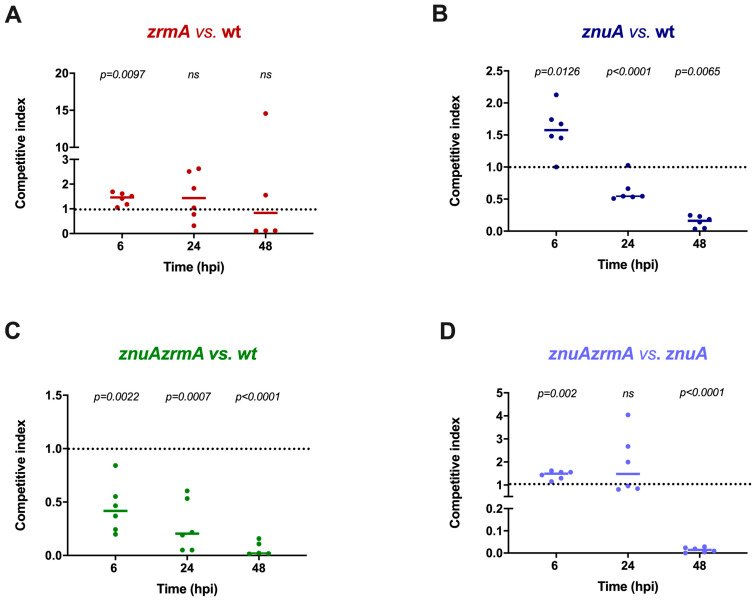
Competition assays of *P. aeruginosa* strains in zebrafish embryos. (**A**) Competition between *zrmA* (strain A) and wild-type (strain B). (**B**) Competition between *znuA* (strain A) and wild-type (strain B). (**C**) Competition between *znuAzrmA* (strain A) and wild-type (strain B). (**D**) Competition between *znuAzrmA* (strain A) and *znuA* (strain B). Each data point represents the competitive index calculated, as described in Section 4.3, on a homogenate of 6 pooled animals. A horizontal straight line indicates the median CI value calculated on the CI values of independent experiments. Statistical analyses were performed by the Student’s *t*-test (*ns*, not significant).

**Figure 3 ijms-24-00944-f003:**
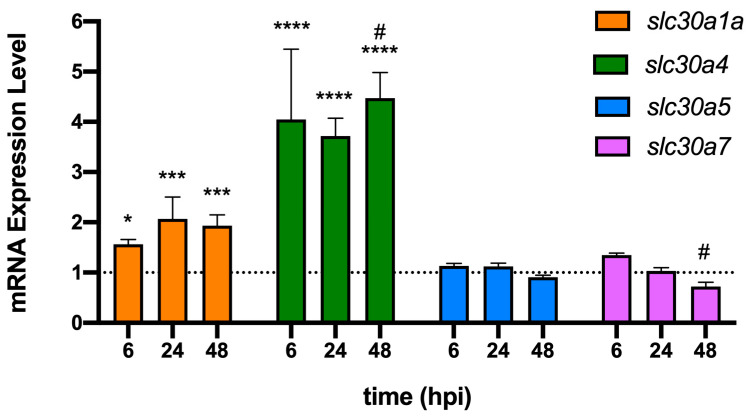
Expression levels of slc30 genes from zebrafish infected with *P. aeruginosa*. RT-PCR was performed on mRNA extracted from infected zebrafish at 6, 24, and 48 hpi. The dotted line at Y = 1 is the expression level of the genes in uninfected animals (control). Data are means ± SD of three independent experiments, and statistical analyses were carried out by two-way ANOVA and Tukey’s multiple comparison test. Asterisks show statistical differences versus uninfected zebrafish (**** *p* < 0.0001; *** *p* < 0.0005; * *p* < 0.05), and hash signs show statistical differences between 24 hpi and 48 hpi for *slc30a4*, and between 6 hpi and 48 hpi for *slc30a7* (# *p* < 0.05).

**Figure 4 ijms-24-00944-f004:**
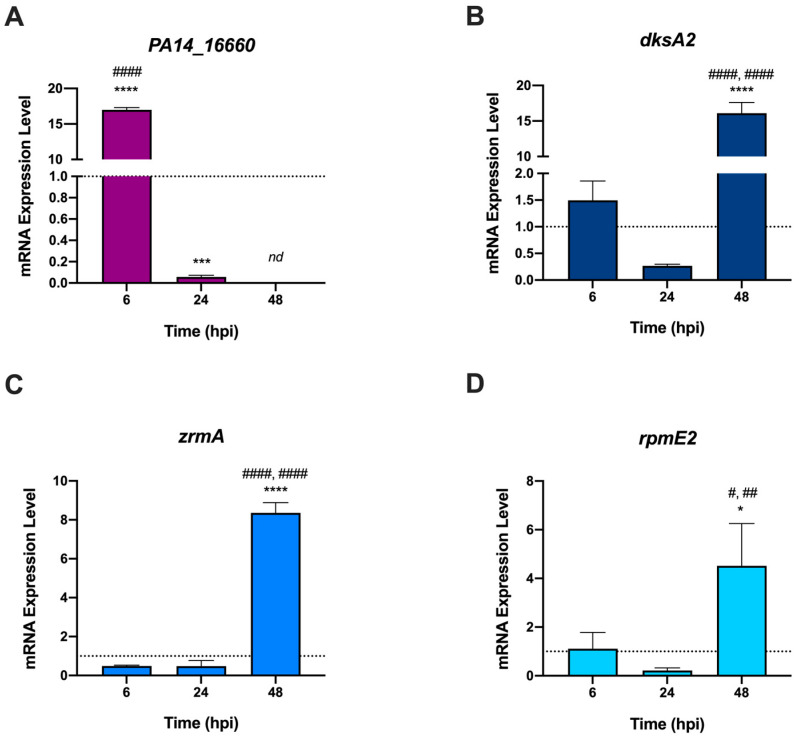
Expression levels of *P. aeruginosa* genes from infected zebrafish. RT-PCR was performed on bacterial mRNA extracted from zebrafish at 6, 24, and 48 hpi. The dotted line at Y = 1 represents the expression level of the genes in bacteria grown in LB medium (control). Data are means ± SD of three independent experiments, and statistical analyses were performed by ordinary one-way ANOVA and Bonferroni’s multiple comparison test. Asterisks show statistical differences versus bacteria grown in LB medium (**** *p* < 0.0001; *** *p* < 0.0005; * *p* < 0.05), hash signs show statistical differences between samples at the different hpi (#### *p* < 0.0001; ## *p* < 0.001; # *p* < 0.05); nd: not detected.

**Table 1 ijms-24-00944-t001:** *P. aeruginosa* strains used in this work.

**Name**	**Strain**	**Relevant Genotype**	**Reference**
wild-type	PA14		Lab collection
*znuA*	MDO101	*znuA*::Gm	[6]
*zrmA*	MDO111	*zrmA*::Gm	[5]
*znuAzrmA*	MDO113	*znuA*::scar *zrmA*::Gm	[5]

**Table 2 ijms-24-00944-t002:** Primers used for RT-PCR.

**Gene**	**Forward (5′-3′)**	**Reverse (5′-3′)**
*dksA2*	AAGCCCAGCAGGACTTCTTC	TGTCGAGCAGCTTCTTCTCC
*ef1α*	TTGAGAAGAAAATCGGTGGTGCTG	GGAACGGTGTGATTGAGGGAAATTC
*oprI*	ATTCTCTGCTCTGGCTCTGG	CGGTCTGCTGAGCTTTCTG
*PA14_16660*	CATCAACGCCCTGATGAGTA	GACTTCGCCTCGATCAACTC
*rpmE2*	GCCGACGTGTACTTCCTGAT	GCGTCACGTAGGGATAGGTC
*rpoD*	CATCGCCAAGAAGTACACCA	CCACGACGGTATTCGAACTT
*slc30a1a*	ACACAAGAACGGGAAGGTCC	GTGACACATATCGGCAGCCT
*slc30a4*	TCCTCGATGTCGGGTCTGAT	GCAGATCCTCCGAGAAGTCG
*slc30a5*	ACTCTGTGGACCACTCAGGA	GCACCCCTCGTCTTAGAAGG
*slc30a7*	TGATGATCGCAGACCCCATC	GCATGGTCCAGAGAAGGAGG
*zrmA*	GACACCCGTATCGAGGACAT	GAAGCCACGGACGTTGTACT

## Data Availability

Not applicable.

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
