# Peer review of "Localized Infections with *P. aeruginosa* Strains Defective in Zinc Uptake Reveal That Zebrafish Embryos Recapitulate Nutritional Immunity Responses of Higher Eukaryotes"

_ijms, 2023, doi:10.3390/ijms24020944_

Round 1

Reviewer 1 Report

This work is very important and very informative. 

1) There is a sentence in Lines 46-47 that needs to be rephrased because it is not clear.  

2) The mutants types need to be more adequately described in the materials and methods section because currently it is not clear what each mutant represent (e.g. what gene is missing and what is its effect a Zn transporter or efflux. 

3) The competitive assay competitive index needs to be explained in terms what the high and low index means.

4) The figures need to be more clearly described in the legend or the results e.g. fig 2b: which strain is higher on the competition index and what that mean! In line 109-110 states that " we observed a clear advantage of the zunaA mutant at 48h post-infection" but this was not explained as of an advantage over what? and which part of the figure you are referring to B or D?

Author Response

This work is very important and very informative.

1) There is a sentence in Lines 46-47 that needs to be rephrased because it is not clear.

OK, the sentence has been modified

2) The mutants types need to be more adequately described in the materials and methods section because currently it is not clear what each mutant represent (e.g. what gene is missing and what is its effect a Zn transporter or efflux.

  1. A sentence has been added to Section 4.1 to explicit the type of mutations introduced. Additional details may be found in the papers cited in Table 1.

3) The competitive assay competitive index needs to be explained in terms what the high and low index means.

  1. A few sentences have been added in Section 4.3

4) The figures need to be more clearly described in the legend or the results e.g. fig 2b: which strain is higher on the competition index and what that mean! In line 109-110 states that " we observed a clear advantage of the zunaA mutant at 48h post- infection" but this was not explained as of an advantage over what? and which part of the figure you are referring to B or D?

  1. The sentence in lines 109-110 has been changed, and the legend of figure 2 has been detailed.

Reviewer 2 Report

General comments

This is an interesting study that can be applied to study host-pathogen interaction regarding metal distribution in the host during infection and pave the road to studying other metals, such as copper, manganese, and iron. However, it needs to be clarified why transcription of slc30a 1a and slc30a4 is high at 48 hpi. It would be expected to see downregulation of those genes at 48 hpi since P. aeruginosa mutants showed growth defects at 48 hpi. If those genes are upregulated at 48 hpi, it will lead to the accumulation of Zn in phagocytes and enhance the replication of the mutants. It would be interesting to monitor zinc localization during P. aeruginosa to determine the restricted-mineral niches in zebrafish. 

Specific comments

Line 52: the sentence is missing a reference.

Line 82: What does hpf stand for?

Line 91 (figure legend): the route of infection is not mentioned 

Line 98: spell out PC since this is the first time it is mentioned

Line 120: Why is this line typed in bold

Line 156: P. aeruginosa must be italicized

Line 257: S. typhimurium must be italicized, and the t of typhimurium must be in lowercase

Author Response

This is an interesting study that can be applied to study host- pathogen interaction regarding metal distribution in the host during infection and pave the road to studying other metals, such as copper, manganese, and iron. However, it needs to be clarified why transcription of slc30a 1a and slc30a4 is high at 48 hpi. It would be expected to see downregulation of those genes at 48 hpi since P. aeruginosa mutants showed growth defects at 48 hpi. If those genes are upregulated at 48 hpi, it will lead to the accumulation of Zn in phagocytes and enhance the replication of the mutants. It would be interesting to monitor zinc localization during P. aeruginosa to determine the restricted-mineral niches in zebrafish.

Reply: The referee wonders why, at 48 hours post-infection, there is not a decrease in the expression of the slc30a1a and slc30a4 genes. In fact, all the experimental data (reduced survival of mutants, higher expression of Zur-regulated genes, reduced expression of the bacterial zinc exporter) indicate that residual bacteria are not localized in macrophages at this time point. We thank the reviewer for giving us the opportunity to comment on this experimental observation, which, in our opinion, provides a further suggestion for the conservation of analogous response mechanisms to infection between zebrafish and higher mammals. Previous studies have in fact demonstrated that slc30a1a and slc30a4 are induced by LPS in human monocyte-derived macrophages, in mouse bone marrow‐derived macrophages and in murine dendritic cells. More generally, it is known that LPS, a component of the membrane of Gram-negative bacteria, can induce a general redistribution of zinc in mammals, which includes, in addition to the accumulation of zinc in macrophages, the decrease of plasma zinc concentration and the accumulation of the metal in the liver, in a form sequestered by metallothionein. Even if the data reported in the work indicate that also in zebrafish macrophages play an important role in the defense against infections, especially in the first phase of the infection, at 48 hpf the infection has not yet resolved, and it is therefore very likely that the bacteria still present in the body provide signals (LPS) that keep

elevated the ability of macrophages to kill bacteria. This is now commented in the discussion, and additional references to previous studies have been included.

We are also very interested in monitoring changes in zinc localization during P. aeruginosa infection, but this will be the subject of future studies that require different experimental approaches

Specific comments

Line 52: the sentence is missing a reference.

Ok, references have been added

Line 82: What does hpf stand for?

  1. A specification has been added in line 82.

Line 91 (figure legend): the route of infection is not mentioned

  1. Now it is mentioned that microinjections were performed in the pericardial cavity as described in Section 4.2.

Line 98: spell out PC since this is the first time it is mentioned

  1. Done.

Line 120: Why is this line typed in bold

  1. The formatting mistake has been corrected.

Line 156: P. aeruginosa must be italicized

  1. Done.

Line 257: S. typhimurium must be italicized, and the t of typhimurium must be in lowercase

Typhimurium is a serovar of Salmonella enterica species. According to the current nomenclature, the name of the strain is capitalized and not italicized. However, the name of the species has been added for clarity in the text.